biomechanics/mechanical engineering

human gait, system identification, spring-loaded inverted pendulum, control, dynamics

**Author for correspondence:**
Anne E. Martin
e-mail: aem34@psu.edu

# Comparing system identification techniques for identifying human-like walking controllers

## Dave Schmitthenner and Anne E. Martin

Penn State, Mechanical Engineering, University Park, PA, USA

 AEM, 0000-0002-2843-6928

While human walking has been well studied, the exact controller is unknown. This paper used human experimental walking data and system identification techniques to infer a human-like controller for a spring-loaded inverted pendulum (SLIP) model. Because the best system identification technique is unknown, three methods were used and compared. First, a linear system was found using ordinary least squares. A second linear system was found that both encoded the linearized SLIP model and matched the first linear system as closely as possible. A third nonlinear system used sparse identification of nonlinear dynamics (SINDY). When directly mapping states from the start to the end of a step, all three methods were accurate, with errors below 10% of the mean experimental values in most cases. When using the controllers in simulation, the errors were significantly higher but remained below 10% for all but one state. Thus, all three system identification methods generated accurate system models. Somewhat surprisingly, the linearized system was the most accurate, followed closely by SINDY. This suggests that nonlinear system identification techniques are not needed when finding a discrete human gait controller, at least for unperturbed walking. It may also suggest that human control of normal, unperturbed walking is approximately linear.

## 1. Introduction

While most people can easily walk without conscious thought, it requires appropriate coordination of leg joints, which arises from the central nervous system [1,2] and/or the natural dynamics of the system [3]. Given the redundancy in the human body, gait control is often assumed to follow a hierarchical structure such as the template and anchor paradigm [4]. Even with a simple template model, there are still many possible gaits, each of which can be described by a set of control parameters. The

**Figure 1.** Typical SLIP model, shown during the double support period. The plus symbol represents the properties of the new stance leg and the minus symbols represent properties of the previous stance leg.

process that dictates how these control parameters are chosen for an imminent step is the controller. Given that the central nervous system is largely a black box, its exact controller is unknown. However, its controller can be inferred by examining the resulting gait. Thus, one objective of this work is to infer a human-like controller given a set of human experimental data.

To accomplish this, the first task is to choose an appropriate template. For walking, there are two common template models—the inverted pendulum model [5] and the spring-loaded inverted pendulum (SLIP) model [6–8]. This work focuses on the SLIP model, partly because previous work has inferred a human-like controller for the inverted pendulum model [9,10]. The SLIP model has two massless, springy legs connected at the hip (figure 1) [6,7]. The hip contains all of the model's mass. For walking, the model alternates between a single support period when only one leg is on the ground and a double support period when both legs are on the ground. This model has two control parameters, leg stiffness and touchdown angle, both of which need to be controlled appropriately to achieve a desired motion. In an ideal case, a single choice of leg stiffness and touchdown angle produces locally stable, periodic walking. However, human gait is only approximately periodic [11], with small perturbations and variations occurring every step. Thus, the controller must adjust the control parameters every step [12].

While a human-like SLIP controller has not yet been identified for walking, many studies have investigated the relationships between the control parameters and walking performance. For humans, leg stiffness tends to increase as walking speed increases, until the transition to running [8,13]. In modelling studies, changing leg stiffness [14] or touchdown angle [7] drastically alters the shape of the ground reaction force (GRF) profile during unperturbed, periodic walking. For perturbed walking, step-to-step leg stiffness adjustment is critical for a stable and efficient gait; for a given biped state, there is a limited stiffness range that allows the next step to be completed [12,15,16]. The range of permitted stiffnesses is primarily determined by walking speed and touchdown angle [7,16]. It is also possible to develop controllers for the SLIP model without considering how human-like the resulting gait is. These studies typically solve the dynamics of the SLIP model, then derive a controller to create a closed system model [17–20]. Beyond this, work with both the inverted pendulum and SLIP models have shown that different controllers can drastically change important gait properties [14,21]. Thus, it is clear that developing an appropriate controller is critical to producing a desired gait.

One way to infer the human controller is to use system identification methods. To do so, both system states (controller input) and observable control parameters (controller output) are measured. These data are then used to find the transformation from the input to the output that best represents the black-box transformation occurring in the central nervous system. While traditionally used for engineering systems, system identification methods have been used to infer neural controllers for both animals [22,23] and humans [9,10,24,25]. Most relevant to this work, system identification has been used to infer a human-like linear controller for a three-dimensional inverted pendulum model [9,10]. This model had two rigid legs connected by a hip joint and transitioned between steps using an instantaneous double-support period. The continuous-time system was converted into a discrete system by taking a Poincaré section at mid-stance. To find the controller, the system states were hip position and velocity at mid-stance, and the control parameters were foot placement and push off impulse. Using an ordinary least-squares method, the human data were converted into a linear system model that

directly transformed the mid-stance state to the next step's mid-stance state. This model was then used to find a linear controller that transformed the mid-stance state into the control parameters. Walking data both with [10] and without [9] external perturbations were used; the results were similar in both cases.

The work in [9,10] treated biped walking as a discrete linear system, which may underestimate nonlinear effects. There has been considerable work showing that gait has nonlinear properties, even when analysed as a discrete system [11,26,27]. Thus, nonlinear system identification techniques may be more appropriate. One method for nonlinear system identification is sparse identification of nonlinear dynamics (SINDY) [28,29]. This method uses a library of nonlinear functions of the system states. It finds a linear combination of nonlinear functions that best models the nonlinear system using sparse identification and experimental data. While SINDY has accurately modelled and controlled many systems, it has not yet been applied to walking. Thus, the second objective of this work was to apply SINDY to walking data and compare the accuracy of SINDY with the simpler and more standard linear system identification techniques used by [9,10].

Therefore, to both (i) infer a human-like controller given a set of human experimental data and (ii) compare different system identification techniques, human walking data were used to identify SLIP-based controllers using three methods. For the first method, a linear Poincaré return map and controller were found using ordinary least squares. A second linear Poincaré return map and controller were found that both encoded the linearized SLIP model and matched the first linear Poincaré return map and controller as closely as possible. A third nonlinear Poincaré return map and controller were found by applying SINDY to the same data. For convenience, the Poincaré return map will be called a model for the remainder of the paper. The results were evaluated using a separate set of human data.

# 2. Methods

As just discussed, three methods were used to extract a system model, $J$, and a controller, $K$, from human data. The first method performed ordinary least-squares analysis on human walking and leg stiffness data to identify a system model and controller, referred to as the linearized system $J_L$ and controller $K_L$ [9]. The second method linearized the SLIP model dynamics and then used the linearized dynamics to optimize a new controller, referred to as the optimized system $J_O$ and controller $K_O$ [10]. The third method used SINDY with control and thus used a library of nonlinear functions to identify a linear combination of nonlinear functions that formed the system model and controller, referred to as the SINDY system $J_S$ and controller $K_S$ [28,29]. The three methods were then compared with each other.

## 2.1. Human data

Walking data from 12 adult subjects (three male, nine female, 18–63 years of age, mass $69 \pm 12$ kg, leg length $0.928 \pm 0.053$ m) were recorded using a VICON motion capture system. Prior to data collection, institutional review board approval from Penn State and informed consent from subjects was obtained. During the experiment, subjects walked on a split-belt instrumented treadmill at $1.1$ m s$^{-1}$ for 6 min, resulting in about 600 steps per subject. Ten per cent of the steps from each subject (approx. 60 steps each or 707 steps total) were held out for evaluation purposes. The remaining steps were used for system identification.

## 2.2. SLIP model

The SLIP model contains a point mass at the hip supported by two legs of different stiffnesses (figure 1). A brief overview of the model is given here; the reader is referred to [6,30] for detailed mathematical derivations. The hip had mass $m$, which was equal to the mass of the subject. Uncompressed leg length was $l_0$, which was equal to the subject's leg length. Hip position $(x, y)$ was measured from the hip to the centre of pressure under the foot, both of which were known directly from the measured experimental data. The GRF for each leg was $F_{leg}$, which was also measured directly. To convert the measured human data into the model's leg stiffness $k$, the effective stiffness at each instant was calculated using

$$k_{inst} = \frac{\|F_{leg}\|}{\Delta l},$$

(2.1)

**Table 1.** Unit-normalized state and control input means and standard deviations, averaged across subjects. $\alpha_{TD}$ is in radians. The hip position is $(x, y)$, the leg stiffness is $k$, and the leg angle at heel contact is $\alpha_{TD}$.

| state/input | mean | std. dev. |
|---|---|---|
| $\dot{x}$ | 0.321 | 0.030 |
| $y$ | 0.973 | 0.077 |
| $\dot{y}$ | 0.027 | 0.011 |
| $k$ | 28.32 | 8.45 |
| $\alpha_{TD}$ | 1.245 | 0.104 |

where $\Delta l$ was the stance leg displacement, which was the difference between the subject's leg length $l_0$ and the distance from the hip to the centre of pressure under the foot. The stiffness was then averaged over the step to find a step stiffness $k$. The stiffness of each leg was held constant throughout a step and changed during that leg's swing phase. The touchdown angle $\alpha_{TD}$ was found by calculating the angle between the ground and the line from hip to centre of pressure at heel contact. A step began at touchdown and ended when the foot lifted off the ground. This model can be mathematically described using a set of continuous-time differential equations [30].

To convert the continuous-time SLIP model into a discrete model, a Poincaré section was taken at vertical leg orientation (VLO). VLO occurs when the centre of mass passes over the stance foot. The discrete system's states were defined as

$$Q^- = \begin{bmatrix} \dot{x}^- \\ y^- \\ \dot{y}^- \end{bmatrix}, \tag{2.2}$$

where $Q$ is the state vector, the superscript $^-$ represents VLO of the current step, and the superscript $^+$ represents VLO of the next step. Since $x$ is measured relative to the stance foot, $x$ is always zero at the Poincaré section since it is taken at VLO. Therefore, $x$ is not considered a system state. The control inputs for a step were

$$U^- = \begin{bmatrix} k^- \\ \alpha_{TD}^- \end{bmatrix}, \tag{2.3}$$

where $U^-$ is the control input vector at the current step. To account for subjects of different sizes, $Q$ and $U$ were normalized using subject leg length $l_0$ and mass $m$. Specifically, hip position was normalized by $l_0$, hip velocity was normalized by $\sqrt{gl_0}$, and stiffness was normalized by $l_0/(mg)$, where $g$ was the acceleration of gravity. The system states and control inputs were calculated for each step of each subject.

## 2.3. Linearized system identification

Once the system states and inputs were known, the next task was to perform system identification to find both a model and a controller. In all cases, the goal was to find a mapping such that given a hip state $Q^-$ and current control input $U^-$, the hip state at the next step $Q^+$ could be predicted. The first method was simple linear system identification. This has been done previously using a stiff-legged three-dimensional model [9]. That model used the same form for $Q$, but used foot placement and push off impulse for $U$, instead of stiffness and touchdown angle. Thus, the desired system for this work was

$$\Delta Q^+ = J_L \, \Delta Q_{aug}^- \tag{2.4}$$

where

$$\Delta Q_{aug}^- = \begin{bmatrix} \Delta Q^- \\ \Delta U^- \end{bmatrix}, \tag{2.5}$$

$\Delta Q^- = Q^- - \bar{Q}$ and $\Delta U^- = U^- - \bar{U}$. For each subject, $\bar{Q}$ and $\bar{U}$ were the mean of the states and control input, averaged over all their steps [9] (table 1). Thus, $J_L$ represented the Jacobian of the linearized system. Additionally, a controller was desired such that, given hip state $Q^-$ and current control input $U^-$, the

control inputs for the next step could be calculated

$$\Delta U^+ = K_L \, \Delta Q^-_{aug},$$ (2.6)

where $K_L$ was the controller for the linearized system.

In order to use the data to find $J_L$ and $K_L$, the Poincaré sections were arranged into matrices such that each column in $\Delta Q^-$ and $\Delta U^-$ represented one step, and the columns of $\Delta Q^+$ and $\Delta U^+$ were the corresponding next step. Using ordinary least squares, $J_L$ and $K_L$ were found using

$$J_L = \left( \left( \begin{bmatrix} \Delta Q^- \\ \Delta U^- \end{bmatrix}^{\mathrm{T}} \right)^{\dagger} [\Delta Q^+]^{\mathrm{T}} \right)^{\mathrm{T}}$$ (2.7)

and

$$K_L = \left( \left( \begin{bmatrix} \Delta Q^- \\ \Delta U^- \end{bmatrix}^{\mathrm{T}} \right)^{\dagger} [\Delta U^+]^{\mathrm{T}} \right)^{\mathrm{T}},$$ (2.8)

where $^{\dagger}$ indicates the Moore–Penrose pseudo-inverse [31]. This calculation used all the training steps for one subject to find $J_L$ and $K_L$ such that equations (2.4) and (2.6) held (approximately) true. A different $J_L$ and $K_L$ were found for each subject based on their step data. Thus, the linearized model and controller were given by equations (2.4) and (2.6). This was the simplest system identification method, but it did not explicitly include the SLIP model.

## 2.4. Optimized system identification

By contrast, the optimized system identification method explicitly included the SLIP model when finding $J$ and $K$ while still resulting in a final linear system. Similar to the previous method, the Poincaré section was taken at VLO, and the discrete model and controller had the same form, using states at the previous step to anticipate and control the next step

$$\Delta Q^+ = J_O \, \Delta Q^-_{aug}$$ (2.9)

and

$$\Delta U^+ = K_O \, \Delta Q^-_{aug}.$$ (2.10)

Equation (2.9) was further broken down into

$$\Delta Q^+ = A \, \Delta Q^-_{aug} + B\Delta U^+,$$ (2.11)

where $J_O = (A + B \, K_O)$. $A$ and $B$ were Jacobians that determined how the states at the next step were affected by small perturbations to previous states and control inputs. Here, all terms were found for each subject separately.

Finding $A$ and $B$ first required finding a periodic walking gait for the SLIP model using optimization. The optimization was performed using the Matlab optimization function fmincon. For each optimization function evaluation, the continuous-time SLIP model was simulated starting at VLO and ending at the next VLO; the objective function was the difference between the beginning and ending states. Both the state vector (giving initial conditions for the SLIP simulation) and the control vector were optimized together to find a periodic gait. The initial conditions for each optimization were $\bar{Q}$ and $\bar{U}$ for that subject, resulting in periodic states closest to the subject's average state. The optimizations were constrained to ensure that the hip did not begin higher than the length of the subject's leg, and to ensure the control inputs did not fall outside of the ranges used by the subject. Once a periodic gait was found, each state was perturbed and simulated for one step, and the differences in states at the next step were used to calculate the Jacobians $A$ and $B$.

Once $A$ and $B$ were known, the next task was to find a controller $K_O$ such that

$$J_O = (A + B \, K_O)$$ (2.12)

was an accurate linear system model [10]. To do this, another optimization was performed to choose $K_O$ so that the optimized system matched the linear system as closely as possible. Because $K_O$ appeared in both the model (equation (2.9)) and controller (equation (2.10)), the error in both needed to be considered. The error in the controller was defined as the vector $e_{K,i} = K_{O,i} - K_{L,i}$ for $i = 1 : 10$. Similarly,

the error in the model was defined as the vector $e_{J,j} = (A_j + B_j K_{O,j}) - J_{L,j}$ for $j = 1 : 15$. These vectors were combined into a single error vector $e = [e_K; e_J]$. Then, the Matlab function fmincon was used to solve the optimization problem

$$\text{minimize}_{K_O} \ \Sigma_{n=1}^{25} \lambda_{\text{err},n} \, e_n. \tag{2.13}$$

$\lambda_{\text{err},n}$ was a vector of scaling parameters for each entry of $J_L$ and $K_L$ that represented uncertainty, calculated as the inverse of the confidence interval of each entry of $K_L$ and $J_L$. These confidence intervals were found via bootstrap statistics, in which $\Delta Q^-$, as well as the corresponding entries in $\Delta U^-$, $\Delta Q^+$, and $\Delta U^+$, were sampled randomly with replacement to find equal length vectors of steps, then $K_L$ and $J_L$ were calculated again with the sampled steps using equations (2.7) and (2.8). This process was repeated 1000 times, and the confidence interval for each entry of the bootstrap $K_L$ and $J_L$ was calculated based on this set of 1000 [32–34]. The solution to equation (2.13) gave $K_O$ while equation (2.12) gave $J_O$. Thus, the optimized model and controller were given by equations (2.9) and (2.10) with the terms calculated using the solutions to equations (2.12) and (2.13).

## 2.5. SINDY system identification

The previous methods linearized the system and the controller. However, human walking is a very complex system, and may not be linear outside of very narrow margins. Thus, finding a nonlinear system and controller may better capture human behaviour. To do so SINDY and SINDY with control (SINDYc) were used [28,29], which required a system of form

$$\Delta Q^+ = \Theta J_S \tag{2.14}$$

and

$$\Delta U^+ = \Theta_U K_S, \tag{2.15}$$

where $\Theta = f(\Delta Q^-, \Delta U^-, \Delta U^+)$ and $\Theta_U = f_U(\Delta Q^-, \Delta U^-)$ were libraries of nonlinear functions. Thus, the system model (equation (2.14)) depended on the current states and control parameters, as well as the control parameters for the next step. By contrast, the controller (equation (2.15)) only depended on the current states and control parameters. $J_S$ and $K_S$ were sparse matrices that specified which functions in $\Theta$ and $\Theta_U$ were relevant. System identification required choosing the functions in the library and finding $J_S$ and $K_S$.

The library of functions contained a wide range of nonlinear functions of the states $\Delta Q^-$ and control parameters $\Delta U^-$ such as $(\dot{x}_{\text{hip}}^-)^2$ and $y_{\text{hip}}^- K_{\text{leg}}^-$. A variety of functions were used because it was not clear *a priori* what all of the important relationships were. From previous work, it is known that horizontal velocity is important [7,8,13,16], so that was included. But since other relationships may also be important, functions not previously related to stiffness or touchdown angle were included as well; for example, horizontal velocity times vertical height, which is related to angular momentum. Because SINDY is well able to balance model accuracy and complexity even with a large library of functions, the functions were chosen systematically knowing that many would not be included in the final model. Thus, the library included all first- and second-order polynomial combinations of the states and control inputs. Additionally, these polynomials were combined with sine and cosine functions of the states and inputs. The trigonometric functions were included mainly because the touchdown angle was a control input and was likely to have a trigonometric relationship with the system. It seemed possible that this may also be true for the other states, so trigonometric functions of all the states and control inputs were included. In total, $\Theta$ contained 274 nonlinear functions and $\Theta_U$ contained 131, with the complete libraries given in the electronic supplementary material.

Once the libraries were chosen, the next task was to determine the appropriate weighting for each function. Given a set of data ($\Delta Q^-$, $\Delta U^-$, $\Delta Q^+$, and $\Delta U^+$), matrices $\Theta$ and $\Theta_U$ were constructed such that every row represented a step and every column represented a different nonlinear function, turning the nonlinear problem into a quasi-linear problem. Thus,

$$\Delta Q^+ = \Theta J_S \tag{2.16}$$

and

$$\Delta U^+ = \Theta_U K_S. \tag{2.17}$$

In many cases, including here, the columns of $\Theta$ and $\Theta_U$ must be normalized to ensure that the restricted

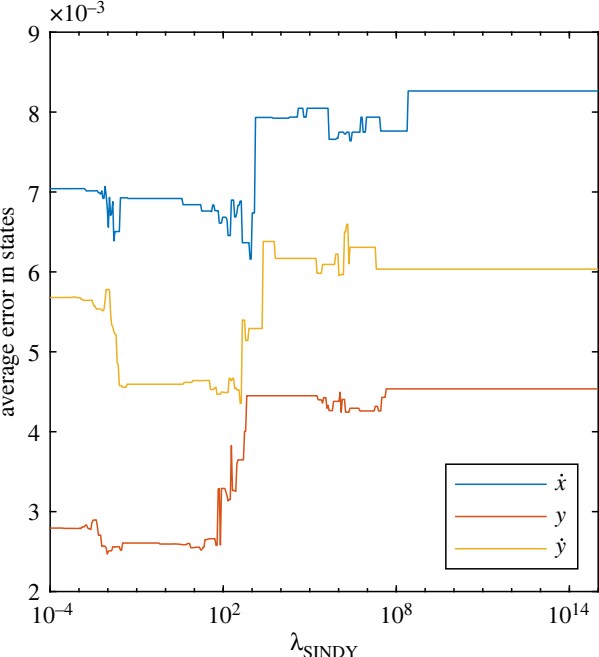

**Figure 2.** Average error in predicting states as a function of the sparsity threshold $\lambda_{\mathrm{SINDY}}$ for a representative subject. Increasing $\lambda_{\mathrm{SINDY}}$ too much causes errors to increase.

isometry property holds [35]; this was done by dividing each column of $\Theta$ and $\Theta_U$ by the norm of that column. Additionally, after $J_S$ and $K_S$ were calculated, each row was divided by the corresponding column norm of $\Theta$ and $\Theta_U$. To solve for $J_S$ and $K_S$, a sequential thresholded least-squares algorithm was used. This algorithm began by solving equations (2.16) and (2.17) using the mldivide function in Matlab

$$J_S = \Theta \backslash \Delta Q^+ \tag{2.18}$$

and

$$K_S = \Theta_U \backslash \Delta U^+ \tag{2.19}$$

which gives the least-squares solution using a QR solver. Then, a sparsity threshold $\lambda_{\mathrm{SINDY}}$ was used to find the indices of $J_S$ and $K_S$ that were too small, meaning that the corresponding columns of $\Theta$ and $\Theta_U$ and the nonlinear functions those columns represent were not important to the system. Those entries were set to zero and the corresponding columns of $\Theta$ and $\Theta_U$ were ignored. The process was repeated iteratively with the remaining indices, comparing the values in $J_S$ and $K_S$ to $\lambda_{\mathrm{SINDY}}$ each time until $J_S$ or $K_S$ converged. This generally took less than 10 iterations. Similar to the other methods, a separate $J_S$ and $K_S$ were found for each subject. Thus, the SINDY model and controller were given by equations (2.14) and (2.15) with $J_S$ and $K_S$ found using the iterative procedure just described.

Each state and control input had a different $\lambda_{\mathrm{SINDY}}$ associated with it because the order of magnitude for each state or control input used could be different. Therefore, $\lambda_{\mathrm{SINDY}}$ for each column of $J_S$ and $K_S$ was found by sweeping through values from $10^{-4}$ to $10^{15}$, evenly spaced in log coordinates. To determine the appropriate value, the resulting $J_S$ and $K_S$ matrices from this sweep were used to predict steps for the held-out data, and the error between the actual and predicted states was calculated for each subject. Section 2.6 describes the procedure used in more detail. In most cases, there was a clear inflection point where higher $\lambda_{\mathrm{SINDY}}$ values caused the error to increase substantially (figure 2). The values just before this point were recorded for each subject, and the logarithmic mean over all of the subjects was used as the final $\lambda_{\mathrm{SINDY}}$ value (table 2).

## 2.6. Evaluation

In order to evaluate how well the different methods identified a human-like system, the models and controllers were evaluated using the held-out data (§2.1). Several different tests were performed. First, model accuracy was evaluated. For each subject and every test step, the experimental states $\Delta Q_{\mathrm{test}}^-$ and

**Table 2.** Sparsity thresholds used.

| associated variable | $\lambda_{SINDY}$ |
| --- | --- |
| $\dot{x}$ | $7.77 \times 10^3$ |
| $y$ | $30.6$ |
| $\dot{y}$ | $9.14 \times 10^4$ |
| $k$ | $9.58 \times 10^4$ |
| $\alpha_{TD}$ | $1.70 \times 10^2$ |

control input $\Delta U_{test}^-$ were used with the linearized model (equation (2.4)), optimized model (equation (2.9)) and SINDY model (equation (2.14)) to predict the states at the next step. For the SINDY model, the output from the controller (equation (2.15)) was used as well because $\Theta$ explicitly includes the control inputs at the next step. ($J_L$ and $J_O$ incorporate their corresponding controllers implicitly.) The predicted states were then compared with experimental data.

The same test data were used with equations (2.6), (2.10) and (2.15) to evaluate the performance of the identified controllers. The calculated control outputs from each step were then compared with experimental data to determine how well the identified controllers matched the human controller.

Finally, a one-step, continuous-time SLIP simulation was performed for each test step. The experimental states $\Delta Q_{test}^-$ and current control parameters $\Delta U_{test}^-$ were used as the initial conditions for the simulation. The touchdown angle and new leg stiffness were determined using equation (2.6), (2.10) or (2.15) depending on the model being tested. In some cases, the walker fell over during the simulation; the number of these falls was recorded. For steps that were completed, the simulated and experimental hip states were compared.

Before calculating summary statistics, all of the test results from all subjects were grouped together and outliers were removed using the Matlab function rmoutliers. Outliers were defined using the default threshold of three scaled median absolute deviations. For the linearized system, less than 5% of the test data were removed. For the optimized system, approximately 18% of the test data were removed when testing for accuracy, and approximately 8% were removed when evaluating the simulation results. For SINDY, approximately 8% of the test data were removed when testing for accuracy, and approximately 2% were removed when evaluating the simulation results. The mean (signed) error, the median absolute error, the RMSE and the standard deviation of the signed error were computed for each of the three evaluation tests for each of the three system identification methods for each of the five extended states ($\dot{x}$, $y$, $\dot{y}$, $k$ and $\alpha_{TD}$). The signed error was approximately normal, motivating the use of a mean while the absolute error did not follow a normal distribution, motivating the use of a median. To determine if there was a consistent bias in the results, Student $t$-tests were used to check if the average (signed) error of each state was statistically different from zero. To compare system identification methods, paired Wilcoxon signed rank tests were used with the absolute error to determine if one method produced errors that were statistically lower and thus better captured the experimental data. In all cases, $\alpha = 0.05$.

# 3. Results

## 3.1. System identification results

All three identified system models had mean errors of the order of $10^{-3}$ in normalized units for all three states although approximately half of the cases had statistically significant non-zero bias (table 3 and figure 3). The median absolute error and RMSE were also of the order of $10^{-3}$–$10^{-2}$. (For comparison, the average $\bar{Q}$ and $\bar{U}$ across subjects is given in table 1). This is less than a 10% error for horizontal hip velocity and hip height and a less than 25% error for vertical hip velocity. The large per cent error in vertical hip velocity is primarily due to the fact that experimental vertical hip velocity is very close to zero at VLO. While the optimized system model predicted $\Delta \dot{x}$ with the lowest bias as indicated by the lowest signed error, it had the highest values for the other measures. The linearized system model had the lowest median absolute error and RMSE for $\Delta \dot{x}$ and was significantly better than the other two systems ($p < 0.001$). The linearized and SINDY models predicted $\Delta y$ with similar low

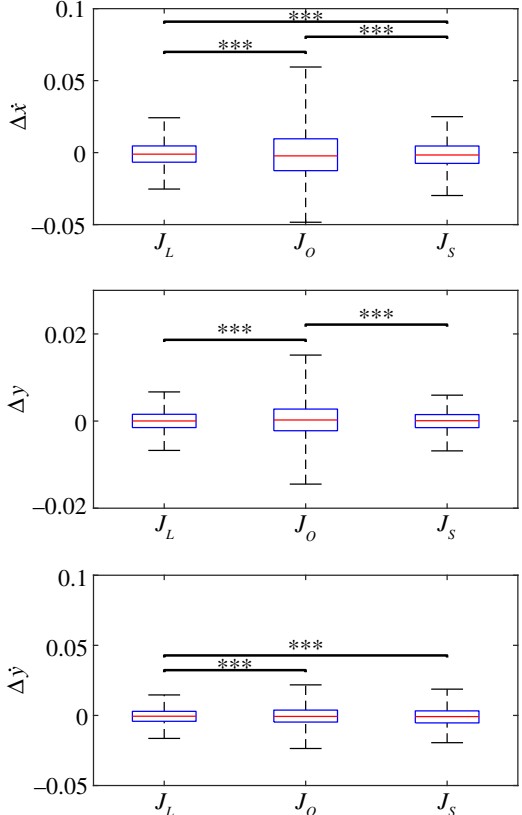

**Figure 3.** Results from testing identified models. In all cases, all three system identification methods were well able to predict the human response. Statistically significant differences between methods are indicated with bars; $^{***}$ indicates $p < 0.001$.

**Table 3.** Mean signed error ($\mu$), median absolute error ($|\mu|$), standard deviation ($\sigma$) and RMSE for identified system models when calculating the next state $\Delta Q^+$ based on $\Delta Q^-$ and $\Delta U^-$. All values are multiplied by $10^3$.

| system | $\Delta \dot{x}^+$ | | | | $\Delta y^+$ | | | | $\Delta \dot{y}^+$ | | | |
|---|---|---|---|---|---|---|---|---|---|---|---|---|
| | $\mu$ | $|\mu|$ | $\sigma$ | RMSE | $\mu$ | $|\mu|$ | $\sigma$ | RMSE | $\mu$ | $|\mu|$ | $\sigma$ | RMSE |
| $J_L$ | −0.93 | 5.58 | 8.60 | 8.60 | −0.08 | 1.53 | 2.40 | 2.40 | −0.52 | 3.65 | 5.50 | 5.50 |
| $J_O$ | −0.56 | 11.4 | 18.0 | 18.0 | 0.35 | 2.52 | 4.20 | 4.20 | 0.01 | 4.30 | 6.40 | 6.40 |
| $J_S$ | −1.40 | 6.12 | 9.30 | 9.40 | −0.10 | 1.50 | 2.40 | 2.40 | −1.10 | 4.31 | 6.50 | 6.60 |

median absolute error and RMSE ($p = 0.75$); both were significantly better than the optimized system model ($p \ll 0.001$). The linearized system model predicted $\Delta \dot{y}$ significantly better than the other two models ($p \ll 0.001$). Therefore, the linearized model had the most accurate results for all three states in general.

The terms in $J_L$ and $J_O$ for all subjects are visualized in the electronic supplementary material. The electronic supplementary material also gives $J_S$ for a representative subject. For the two linear models $J_L$ and $J_O$, high-valued matrix entries correspond to variables that the methods identified as important, while entries that had a consistent sign across subjects indicated that the variable had a consistent effect on gait. For $J_S$, non-zero entries correspond to functions that were important; states or control parameters that have many non-zero functions within the library and across subjects were assumed to be important. For all three system identification methods, entries related to $\Delta \dot{x}$ were typically larger in magnitude, which was expected since speed is known to be correlated with stiffness. Particularly for the linearized model, the signs of the terms across subjects were consistent. Interestingly, large entries related to $\Delta y$ were also prevalent in all three models. Touchdown angle did not have large

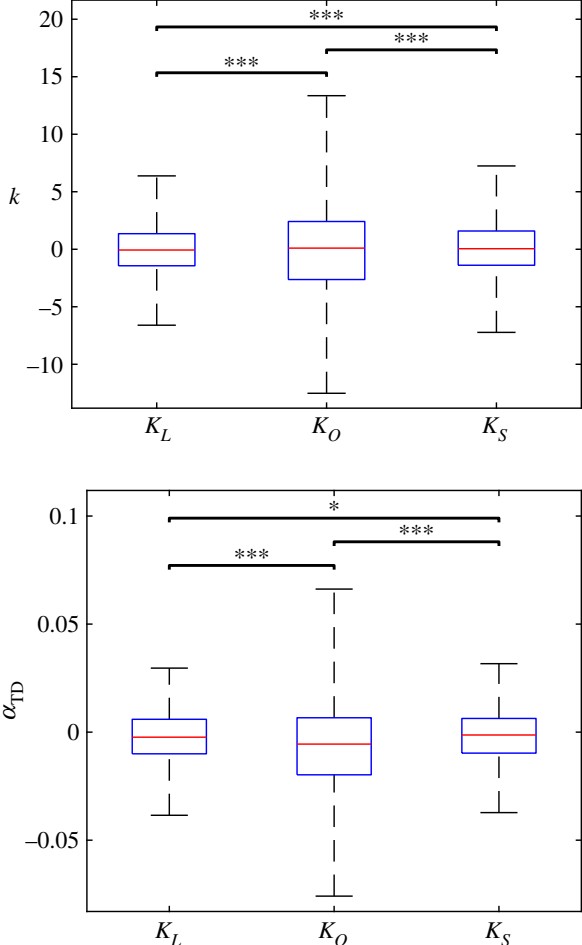

**Figure 4.** Results from testing identified controllers. Similar to the states, the errors were generally low, indicating good agreement between the experimental and predicted control parameters. Statistically significant differences between methods are indicated with bars; $^{*}$ indicates $p < 0.05$ and $^{***}$ indicates $p < 0.001$.

**Table 4.** Mean error ($\mu$), median absolute error ($|\mu|$), standard deviation ($\sigma$), and RMSE for identified controllers when calculating $\Delta U^{+}$ based on $\Delta Q^{-}$ and $\Delta U^{-}$.

| system | $\Delta k$ | | | | $\Delta \alpha_{TD}$ (rad) | | | |
|---|---|---|---|---|---|---|---|---|
| | $\mu$ | $|\mu|$ | $\sigma$ | RMSE | $\mu$ $(10^{-3})$ | $|\mu|$ $(10^{-3})$ | $\sigma$ $(10^{-3})$ | RMSE $(10^{-3})$ |
| $K_L$ | −0.11 | 1.40 | 2.37 | 2.37 | −2.50 | 7.76 | 11.6 | 11.9 |
| $K_O$ | 0.30 | 2.53 | 4.51 | 4.54 | −7.30 | 13.8 | 26.0 | 26.6 |
| $K_S$ | −0.08 | 1.48 | 2.67 | 2.67 | −2.20 | 8.01 | 12.5 | 12.6 |

magnitudes, but the signs of the terms tended to be consistent for the linear models. The matrices calculated with SINDY also tended to have larger entries related to the sine and cosine of $\alpha_{TD}$.

## 3.2. Controller ID results

All of the identified controllers predicted leg stiffness with mean errors below 1 in normalized stiffness units, indicating minimal bias ($p > 0.2$). The median absolute error and RMSE ranged between 5% and 15% of mean leg stiffness (table 4 and figure 4). The linearized system controller mimicked the human stiffness response the best ($p \ll 0.001$) followed by the SINDY controller ($p \ll 0.001$). For touchdown angle, the controllers all had mean signed errors of the order of $10^{-3}$ radians. Despite the small error magnitude, there was consistent bias for all three controllers ($p \ll 0.001$). When considering accuracy,

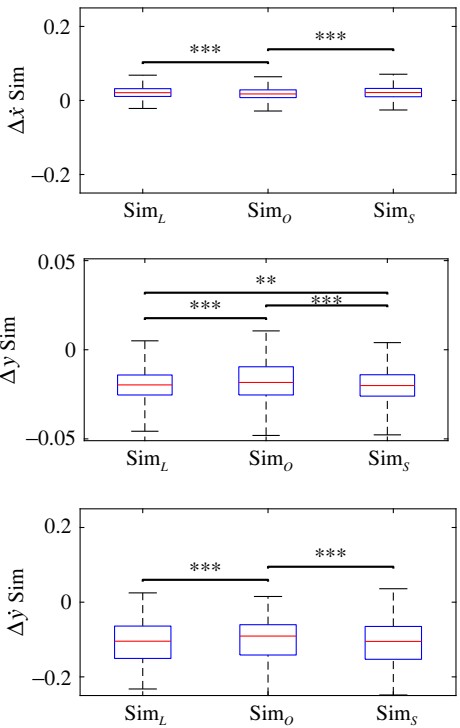

**Figure 5.** Results from using identified controllers in simulation. There was significant bias in the predicted states regardless of which system identification method was used. Nevertheless, the errors in $\dot{x}$ and $y$ were acceptable. Statistically significant differences between methods are indicated with bars; $^{**}$ indicates $p < 0.01$, $^{***}$ indicates $p < 0.001$.

**Table 5.** Mean error ($\mu$), median absolute error ($|\mu|$), standard deviation ($\sigma$), and RMSE for identified controllers when calculating $\Delta U^+$ based on $\Delta Q^-$ and $\Delta U^-$, then simulating for one step using SLIP model to find $\Delta Q^+$. All values are multiplied by $10^3$.

| system | $\Delta \dot{x}^+$ | | | | $\Delta y^+$ | | | | $\Delta \dot{y}^+$ | | | |
|---|---|---|---|---|---|---|---|---|---|---|---|---|
| | $\mu$ | $|\mu|$ | $\sigma$ | RMSE | $\mu$ | $|\mu|$ | $\sigma$ | RMSE | $\mu$ | $|\mu|$ | $\sigma$ | RMSE |
| $J_L$ | 21.7 | 21.1 | 14.9 | 26.3 | −20.4 | 19.7 | 9.4 | 22.5 | −109.2 | 105 | 55.5 | 123 |
| $J_O$ | 17.4 | 18.0 | 15.7 | 23.4 | −16.8 | 18.4 | 11.0 | 20.1 | −99.4 | 90.8 | 59.1 | 116 |
| $J_S$ | 21.5 | 21.4 | 15.2 | 26.3 | −20.6 | 20.1 | 10.0 | 22.9 | −110.5 | 105 | 56.5 | 124 |

the median absolute error and RMSE were less than 3% of the mean experimental values. Similar to stiffness, the linearized controller predicted the human response best ($p < 0.03$) followed closely by the SINDY controller. The optimized controller was by far the least accurate ($p \ll 0.001$). Therefore, the linearized model had the most accurate controller in general.

The terms in $K_L$ and $K_O$ for all subjects are visualized in the electronic supplementary material. The electronic supplementary material also gives $K_S$ for a representative subject. Similar to the model results, entries related to $\Delta \dot{x}$ and $\Delta y$ were typically larger in magnitude for all three models, although the sign of the terms varied by subject. For the linearized controller, the new touchdown angle was also strongly influenced by the previous touchdown angle. For SINDY, $\alpha_{\text{TD}}$ had the most non-zero functions although all three states had large numbers of non-zero functions as well.

## 3.3. Simulation results

When the controllers (equations (2.6), (2.10) and (2.15)) were used in conjunction with a continuous-time SLIP model simulation, all three states for all three controllers had significant bias ($p \ll 0.001$, table 5 and figure 5). Horizontal hip velocity at VLO was consistently overestimated by 5–10% on average. Vertical hip velocity at VLO was consistently underestimated; the magnitude of the error was similar to the

**Table 6.** Falls recorded during simulations.

| controller | subject number | | | | | | | | | | | |
|---|---|---|---|---|---|---|---|---|---|---|---|---|
| | 1 | 2 | 3 | 4 | 5 | 6 | 7 | 8 | 9 | 10 | 11 | 12 |
| $K_L$ | 0 | 0 | 0 | 0 | 0 | 0 | 0 | 0 | 0 | 0 | 0 | 0 |
| $K_O$ | 0 | 10 | 0 | 0 | 0 | 0 | 0 | 0 | 59 | 5 | 0 | 0 |
| $K_S$ | 0 | 1 | 1 | 0 | 0 | 0 | 0 | 0 | 3 | 0 | 0 | 0 |

average experimental value. Regardless of the system identification method, vertical hip velocity had the largest absolute error and largest per cent error of the three states. Vertical hip height at VLO was consistently underestimated by approximately 2% on average. The optimized controller produced steps that matched the experimental states significantly better than both the linearized controller and the SINDY controller ($p < 0.001$); it had the lowest mean signed error, lowest median absolute error and lowest RMSE for all three states. The linearized controller and SINDY controller performed similarly in simulation ($p = 0.01$ for $\Delta y$ and $p > 0.45$ for $\Delta \dot{x}$ and $\Delta \dot{y}$). However, the SLIP model fell during simulation: 0 times for the linearized controller, 74 times for the optimized controller (including 59 times for one subject) and five times for the SINDY controller (table 6). Thus, the linearized and SINDY controllers appear to be more robust but less accurate than the optimized controller when performing continuous-time simulations.

# 4. Discussion

All three system identification methods performed well, with errors below 10% in most cases. Somewhat surprisingly, the linearized system model and controller were the most accurate overall. The SINDY system model and controller were almost, but not quite, as accurate. Given that the linearized model performed best, it appears that nonlinear system identification techniques are not needed when finding a discrete human gait model, at least for unperturbed walking. The optimized controller was generally the least accurate model, although it produced the most accurate results when the controller was used with a continuous-time SLIP model simulation. However, the optimized controller was significantly more likely to choose control parameters that caused a fall in simulation. For the set of data used here, the optimized controller failed completely for one subject and produced significantly more falls for the remaining subjects. Data from these falls were not included in the summary values; if they were, the values for optimized controller would have been much worse.

Regardless of the system identification method, the simulations were far less accurate than directly finding a mapping from one step to the next. When simulated, all three states had significant errors and bias, but the error in $\dot{y}$ was particularly high. In some cases, the hips ended the simulation step moving in the wrong vertical direction. Because the control parameters were generally similar to the human values, but the simulations still performed poorly, this suggests that the SLIP model did not properly capture all human walking dynamics [12]. This may partially explain why the optimized controller was the least accurate method. Because the optimized model directly fit the data to the SLIP model, it was limited to SLIP dynamics. By contrast, the other two methods were informed by the SLIP model but did not directly include it and were therefore able to identify important aspects not present in the SLIP model.

All three models found that horizontal hip velocity was an important factor in predicting at least some of the states and/or control parameters. Previous work using the three-dimensional inverted pendulum models also found that horizontal hip velocity was important [9,10]. Beyond horizontal hip velocity, the system identification methods also seemed to identify $\Delta y$ as an important factor in walking. This may be because $y$ is directly related to how 'compressed' the leg is, which in turn alters the GRF. $\Delta y$ was not an important factor for the three-dimensional inverted pendulum model in [9]. This difference is not particularly surprising because hip height is primarily a measure of how tilted the leg is for an inverted pendulum model. In other words, hip height captures different dynamical information between the two models, so the impact on the system was expected to be different. Hip vertical velocity did not appear to be important for either the linearized or optimized models, and appeared to be slightly less important than horizontal hip velocity for the SINDY model. This is not

entirely surprising because the biped was analysed at VLO when hip height should be close to a maximum for the step, so there should be little vertical velocity. Leg stiffness appeared to be an important factor for the linearized and optimized systems. This was expected because many previous works have shown that leg stiffness alters gait [12,15,16]. Surprisingly, there were very few terms containing stiffness in the SINDY model and controller. Touchdown angle was a consistently important factor for all three system identification methods. While touchdown angle has not been studied as much as stiffness, previous simulation studies have also shown that it is important [7].

Although humans do not completely follow simple continuous time models, their response to normal walking variability appears to be approximately linear. This seems to hold for both the inverted pendulum model [9] and the SLIP model studied in this work. Switching from the very simple linearized system identification method to the nonlinear system identification method SINDY slightly reduced accuracy. This may be because the library of functions used for SINDY did not include some important, but currently unknown, function or measurement, such as angular velocity or angular momentum. It is also possible that SINDY needed more data to find a more accurate model. However, because the linear controller performed well in the first place, it is possible that steady state walking is actually well approximated by a linear approximation of a Poincaré section. Because this work only used unperturbed walking, it is unclear if the linear approximation holds even with noticeable perturbations to walking. It may, since linear system identification found very similar systems for both unperturbed and perturbed walking [9,10]. However, neither of those works compared the linear system with a nonlinear system, and Joshi & Srinivasan [10] noted that there was some indication of nonlinearity. Future work using data that include highly perturbed steps may shed more light on this question. It may also lead to a more robust solution, since the behaviours to correct the gait will be exaggerated and therefore easier for the algorithms to identify.

Nonlinear models may also be required if the control parameters are allowed to vary during a step. The system identification methods used in this work assumed that gait was quasi-periodic and that it could be analysed by taking a Poincaré section. The methods further assumed that stiffness only changed at leg touchdown. This is not entirely realistic because humans can and do modulate their muscles in the middle of a step, which in turn alters the stiffness during the step [36,37]. Thus, a method that identifies a continuous time controller or model may be more appropriate. In particular, such a controller may improve the continuous-time simulation results because it could capture dynamics beyond the basic SLIP dynamics. Finding a continuous time controller for a quasi-periodic system is likely to be prohibitively difficult using linearization techniques, but may be possible through SINDY, which has a continuous time form not used here [29].

In summary, both linear and nonlinear system identification methods can be used to find human-like, discrete-time walking models and controllers based on the SLIP model. At least for the data used here (unperturbed walking from 12 subjects with approx. 600 steps each), a basic linear system identification method worked best, followed closely by the nonlinear system identification method SINDY. Explicitly incorporating the SLIP model into the linear system identification method did not work as well. These results may indicate that the central nervous system uses an approximately linear controller when reacting to small perturbations caused by noise in the neuromuscular system, as opposed to the nonlinear controller that is often assumed.

Data accessibility. This article has no additional data.
Competing interests. We declare we have no competing interests.
Funding. This work was supported by the NSF under award 1727540.
Acknowledgements. The authors thank Dell Snook for his help with data collection.

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
