## [Peer Review File · Royal Society Open Science]

Review History

RSOS-211031.R0 (Original submission)

Review form: Reviewer 1

Is the manuscript scientifically sound in its present form?

Yes

Are the interpretations and conclusions justified by the results?

Yes

Is the language acceptable?

Yes

Do you have any ethical concerns with this paper?

No

Have you any concerns about statistical analyses in this paper?

No

Recommendation?

Major revision is needed (please make suggestions in comments)

Comments to the Author(s)

This manuscript investigates the system identification of human walking with three different models - (linear) ordinary least squares, SLIP model (optimized), and nonlinear SINDY. Comparisons were made among them through model accuracy against human subjects during unperturbed walking to determine the best controller. Errors were calculated at mid-stance, and the model controllers were also used in a gait simulation. The linear system was the most accurate, followed by SINDY, suggesting nonlinear identification techniques might not be necessary to determine unperturbed walking controllers.

The manuscript was generally well-written and clear. Claims and conclusions were supported by the data, and the authors adequately addressed the limitations of their study.

I have a few mostly moderate concerns. My main concerns are about the treatment of human data (missing methodology, ethics statement, and removal of subjects without explanation).

1. Human data:

- a. Please include number of males/females and age range.
 - b. The ethics and informed consent statements were also not included in the manuscript itself.
 - c. The model was compared against human data, but it is unclear how leg stiffness, touchdown angle, and other measures were determined from the human data.
2. In the code, five subjects were skipped "due to poor data," but I didn't find a statement or explanation about their removal in the manuscript. The manuscript should detail all the data taken (the number of subjects is actually 17, not 12) and then a clear explanation why some could not be used.
3. End of introduction: Could the authors be more specific by "...using the SLIP model and informed by the first method." The abstract has the same wording. It is unclear at this point what "informed by the first method" means.
4. Table 1: The variables do not match Figure 1 or the equations that reference the table. Could similar variable names be used for consistency? Please restate what each variable refers to. Did the subjects exhibit a wide range of behavior (e.g. what was the standard deviation)?
5. Line 256 indicates that outliers were removed with `rmoutliers`. What was the criteria for removing the outliers, and how many outliers were removed? Since the summary statistics referred to human data, should the statement be included as part of Section 2.1?
6. I suggest ending the manuscript with a broader statement or two on the implications of the work, especially since the finding (linear performed best over nonlinear) was interesting and surprising.
7. Supplementary data: the leg lengths for some of the subjects are zero. I assume this is related to removal of subjects.

Review form: Reviewer 2 (Vijay Bhaskar Semwal)

Is the manuscript scientifically sound in its present form?

Yes

Are the interpretations and conclusions justified by the results?

Yes

Is the language acceptable?

Yes

Do you have any ethical concerns with this paper?

No

Have you any concerns about statistical analyses in this paper?

No

Recommendation?

Major revision is needed (please make suggestions in comments)

Comments to the Author(s)

The author(s) has presented work on Comparing System Identification Techniques for Identifying 2 Human-like Walking Controllers. The paper has designed SLIP similar controller using human data for gait stability and controllability. However, the approach is simple and straightforward. The methodology and results are partially justified. The major rewrite of paper is required. The author(s) should consider the following review and revised fully.

1. The abstract needs to be written again as it are not clearly explaining the work done
2. The conclusion needs to be aligned with the abstract and future work should be more comprehensive.
3. The title is not matching with the proposed methodology. The title is not suitable rephrase it or change. The question mark cannot be title.
4. On page , RELATED WORKS, you have mentioned only a select few algorithms that were used in human gait and bipedal gait, but in reality, there are numerous Machine learning and Deep learning methods which have been applied in the Human activity and gait recognition space. I suggest going through the literature review once again. Example: "Deep ensemble learning approach for lower extremity activities recognition using wearable sensors" by Jain et al. refer this work.
5. Clearly explain the methodology as to how data pre-processing is performed. Merely mentioning that it is performed is not sufficient.
6. I didn't see any mathematical expressions and theoretical background of the proposed methodology. Only modifying the Deep learning model does not prove the proposed method's novelty.
7. A very vague description is provided for the model description. I suggest backing your statements with some results. Must provide the depth literature review, back ground support and application of proposed work.
8. Proofreading of the paper is required. There are lots of grammatical mistakes which need to be addressed.
9. Must explain the motivation and feasibility in term of cost and applicability.
10. Must compare with state of at work. Author can utilized ELM based classification technique and show the performance in table. Refer and cite this work: Patil, Prithvi, et al. "Clinical human gait classification: extreme learning machine approach." 2019 1st international conference on advances in science, engineering and robotics technology (ICASERT). IEEE, 2019.
12. The workflow diagram is vague can design again.
- 13- The author(s) has missed many latest work of gait in recent year. The work should be compared with following papers in literature review to support and prove your results superiority.

Semwal, Vijay Bhaskar, et al. "Design of vector field for different subphases of gait and regeneration of gait pattern." IEEE Transactions on Automation Science and Engineering 15.1 (2016): 104-110.

Nandi, Gora Chand, et al. "Modeling bipedal locomotion trajectories using hybrid automata." 2016 IEEE region 10 conference (TENCON). IEEE, 2016.

Gupta, Anjali, et al. "Multiple task human gait analysis and identification: ensemble learning approach." *Emotion and information processing*. Springer, Cham, 2020. 185-197.

Bijalwan, Vishwanath, et al. "Fusion of Multi-sensor based Biomechanical Gait Analysis using Vision and Wearable Sensor." *IEEE Sensors Journal* (2021).

Semwal, Vijay Bhaskar, et al. "Pattern identification of different human joints for different human walking styles using inertial measurement unit (IMU) sensor." *Artificial Intelligence Review* (2021): 1-21.

Patil, Prithvi, et al. "Clinical human gait classification: extreme learning machine approach." 2019 1st international conference on advances in science, engineering and robotics technology (ICASERT). IEEE, 2019.

Decision letter (RSOS-211031.R0)

Dear Dr Martin

The Editors assigned to your paper RSOS-211031 "Comparing System Identification Techniques for Identifying Human-like Walking Controllers" have now received comments from reviewers and would like you to revise the paper in accordance with the reviewer comments and any comments from the Editors. Please note this decision does not guarantee eventual acceptance.

Please submit your revised manuscript and required files (see below) no later than 21 days from today's (ie 04-Oct-2021) date. Note: the ScholarOne system will 'lock' if submission of the revision is attempted 21 or more days after the deadline. If you do not think you will be able to meet this deadline please contact the editorial office immediately.

Kind regards,
Royal Society Open Science Editorial Office

on behalf of Dr Jonas Rubenson (Associate Editor) and Kevin Padian (Subject Editor)
openscience@royalsociety.org

Associate Editor Comments to Author (Dr Jonas Rubenson):

Dear Dr. Martin,

As you will see, the reviewers are overall positive about your study and manuscript. Nevertheless, the Reviewers raise some concerns that you will need to address before this work can be considered for publication. Specifically, Reviewer 1 raises some important concerns regarding the description of the human subject research and the treatment of the human data that I agree should be addressed. I hope that these concerns can be addressed in revision.

All the best,
Jonas Rubenson

Reviewer comments to Author:

Reviewer: 1

Comments to the Author(s)

This manuscript investigates the system identification of human walking with three different models - (linear) ordinary least squares, SLIP model (optimized), and nonlinear SINDY. Comparisons were made among them through model accuracy against human subjects during unperturbed walking to determine the best controller. Errors were calculated at mid-stance, and the model controllers were also used in a gait simulation. The linear system was the most accurate, followed by SINDY, suggesting nonlinear identification techniques might not be necessary to determine unperturbed walking controllers.

The manuscript was generally well-written and clear. Claims and conclusions were supported by the data, and the authors adequately addressed the limitations of their study.

I have a few mostly moderate concerns. My main concerns are about the treatment of human data (missing methodology, ethics statement, and removal of subjects without explanation).

1. Human data:

- a. Please include number of males/females and age range.
- b. The ethics and informed consent statements were also not included in the manuscript itself.
- c. The model was compared against human data, but it is unclear how leg stiffness, touchdown angle, and other measures were determined from the human data.

2. In the code, five subjects were skipped "due to poor data," but I didn't find a statement or explanation about their removal in the manuscript. The manuscript should detail all the data taken (the number of subjects is actually 17, not 12) and then a clear explanation why some could not be used.

3. End of introduction: Could the authors be more specific by "...using the SLIP model and informed by the first method." The abstract has the same wording. It is unclear at this point what "informed by the first method" means.

4. Table 1: The variables do not match Figure 1 or the equations that reference the table. Could similar variable names be used for consistency? Please restate what each variable refers to. Did the subjects exhibit a wide range of behavior (e.g. what was the standard deviation)?

5. Line 256 indicates that outliers were removed with `rmoutliers`. What was the criteria for removing the outliers, and how many outliers were removed? Since the summary statistics referred to human data, should the statement be included as part of Section 2.1?

6. I suggest ending the manuscript with a broader statement or two on the implications of the work, especially since the finding (linear performed best over nonlinear) was interesting and surprising.
7. Supplementary data: the leg lengths for some of the subjects are zero. I assume this is related to removal of subjects.

Reviewer: 2

Comments to the Author(s)

The author(s) has presented work on Comparing System Identification Techniques for Identifying 2 Human-like Walking Controllers. The paper has designed SLIP similar controller using human data for gait stability and controllability. However, the approach is simple and straightforward. The methodology and results are partially justified. The major rewrite of paper is required. The author(s) should consider the following review and revised fully.

1. The abstract needs to be written again as it are not clearly explaining the work done
2. The conclusion needs to be aligned with the abstract and future work should be more comprehensive.
3. The title is not matching with the proposed methodology. The title is not suitable rephrase it or change. The question mark cannot be title.
4. On page , RELATED WORKS, you have mentioned only a select few algorithms that were used in human gait and bipedal gait, but in reality, there are numerous Machine learning and Deep learning methods which have been applied in the Human activity and gait recognition space. I suggest going through the literature review once again. Example: "Deep ensemble learning approach for lower extremity activities recognition using wearable sensors" by Jain et al. refer this work.
5. Clearly explain the methodology as to how data pre-processing is performed. Merely mentioning that it is performed is not sufficient.
6. I didn't see any mathematical expressions and theoretical background of the proposed methodology. Only modifying the Deep learning model does not prove the proposed method's novelty.
7. A very vague description is provided for the model description. I suggest backing your statements with some results. Must provide the depth literature review, back ground support and application of proposed work.
8. Proofreading of the paper is required. There are lots of grammatical mistakes which need to be addressed.
9. Must explain the motivation and feasibility in term of cost and applicability.
10. Must compare with state of at work. Author can utilized ELM based classification technique and show the performance in table. Refer and cite this work: Patil, Prithvi, et al. "Clinical human gait classification: extreme learning machine approach." 2019 1st international conference on advances in science, engineering and robotics technology (ICASERT). IEEE, 2019.
12. The workflow diagram is vague can design again.
- 13- The author(s) has missed many latest work of gait in recent year. The work should be compared with following papers in literature review to support and prove your results superiority.

Semwal, Vijay Bhaskar, et al. "Design of vector field for different subphases of gait and regeneration of gait pattern." IEEE Transactions on Automation Science and Engineering 15.1 (2016): 104-110.

Nandi, Gora Chand, et al. "Modeling bipedal locomotion trajectories using hybrid automata." 2016 IEEE region 10 conference (TENCON). IEEE, 2016.

Gupta, Anjali, et al. "Multiple task human gait analysis and identification: ensemble learning approach." Emotion and information processing. Springer, Cham, 2020. 185-197.

Bijalwan, Vishwanath, et al. "Fusion of Multi-sensor based Biomechanical Gait Analysis using Vision and Wearable Sensor." *IEEE Sensors Journal* (2021).

Semwal, Vijay Bhaskar, et al. "Pattern identification of different human joints for different human walking styles using inertial measurement unit (IMU) sensor." *Artificial Intelligence Review* (2021): 1-21.

Patil, Prithvi, et al. "Clinical human gait classification: extreme learning machine approach." 2019 1st international conference on advances in science, engineering and robotics technology (ICASERT). IEEE, 2019.

===PREPARING YOUR MANUSCRIPT===

===PREPARING YOUR REVISION IN SCHOLARONE===

Please ensure that you include a summary of your paper at Step 2 'Type, Title, & Abstract'. This should be no more than 100 words to explain to a non-scientific audience the key findings of your

research. This will be included in a weekly highlights email circulated by the Royal Society press office to national UK, international, and scientific news outlets to promote your work.

Author's Response to Decision Letter for (RSOS-211031.R0)

See Appendix A.

RSOS-211031.R1 (Revision)

Review form: Reviewer 1

Is the manuscript scientifically sound in its present form?

Yes

Are the interpretations and conclusions justified by the results?

Yes

Is the language acceptable?

Yes

Do you have any ethical concerns with this paper?

No

Have you any concerns about statistical analyses in this paper?

No

Recommendation?

Accept as is

Comments to the Author(s)

The authors have addressed all my concerns in their response and updated manuscript.

Decision letter (RSOS-211031.R1)

Dear Dr Martin,

It is a pleasure to accept your manuscript entitled "Comparing System Identification Techniques for Identifying Human-like Walking Controllers" in its current form for publication in Royal Society Open Science. The comments of the reviewer(s) who reviewed your manuscript are included at the foot of this letter.

The proof of your paper will be available for review using the Royal Society online proofing system and you will receive details of how to access this in the near future from our production

office (openscience_proofs@royalsociety.org). We aim to maintain rapid times to publication after acceptance of your manuscript and we would ask you to please contact both the production office and editorial office if you are likely to be away from e-mail contact to minimise delays to publication. If you are going to be away, please nominate a co-author (if available) to manage the proofing process, and ensure they are copied into your email to the journal.

on behalf of Dr Jonas Rubenson (Associate Editor) and Kevin Padian (Subject Editor)
openscience@royalsociety.org

Associate Editor Comments to Author (Dr Jonas Rubenson):

Associate Editor: 1

Comments to the Author:

Dear Dr. Martin,

As you will see, R1 is satisfied with your revised manuscript. I therefore do not see a need for further revisions of this work.

All the best,
Jonas Rubenson

Reviewer comments to Author:

Reviewer: 1

Comments to the Author(s)

The authors have addressed all my concerns in their response and updated manuscript.

Appendix A

Response to reviewers of Royal Society Open Science Manuscript ID RSOS-211031: *Comparing System Identification Techniques for Identifying Human-like Walking Controllers*

The authors thank the reviewers, particularly Reviewer 1, and editor for their comments. Responses to the comments are detailed below, and major revisions to the document are in blue.

Associate Editor Comments:

Dear Dr. Martin,

As you will see, the reviewers are overall positive about your study and manuscript. Nevertheless, the Reviewers raise some concerns that you will need to address before this work can be considered for publication. Specifically, Reviewer 1 raises some important concerns regarding the description of the human subject research and the treatment of the human data that I agree should be addressed. I hope that these concerns can be addressed in revision.

All the best,

Jonas Rubenson

Thank you for your summary of the important points to be addressed. As outlined below, we have added additional details about the human subjects and appropriate ethics to Sec. 2.1. Sec. 2.2 now explains how we converted the experimentally measured data into SLIP model parameters. In the code, we now explain that certain subjects were excluded prior to performing system identification due to data collection or protocol issues.

Reviewer 1 Comments:

This manuscript investigates the system identification of human walking with three different models - (linear) ordinary least squares, SLIP model (optimized), and nonlinear SINDY. Comparisons were made among them through model accuracy against human subjects during unperturbed walking to determine the best controller. Errors were calculated at mid-stance, and the model controllers were also used in a gait simulation. The linear system was the most accurate, followed by SINDY, suggesting nonlinear identification techniques might not be necessary to determine unperturbed walking controllers.

The manuscript was generally well-written and clear. Claims and conclusions were supported by the data, and the authors adequately addressed the limitations of their study.

Thank you.

I have a few mostly moderate concerns. My main concerns are about the treatment of human data (missing methodology, ethics statement, and removal of subjects without explanation).

1. Human data:

a. Please include number of males/females and age range.

We have added the requested information. The sentence now reads *“Walking data from 12 adult subjects (3 male, 9 female, 18 to 63 years of age, mass 69±12 kg, leg length 0.928±0.053 m) was recorded using a VICON motion capture system.”*

b. The ethics and informed consent statements were also not included in the manuscript itself.

We now include the ethics statement in the manuscript itself: *“Prior to data collection, institutional review board approval from Penn State and informed consent from subjects was obtained.”*

c. The model was compared against human data, but it is unclear how leg stiffness, touchdown angle, and other measures were determined from the human data.

This information has been added to the first paragraph of Sec. 2.2. It now reads

The SLIP model contains a point mass at the hip supported by two legs of different stiffnesses (Figure 1). A brief overview of the model is given here; the reader is referred to [6,30] for detailed mathematical derivations. The hip had mass m , which was equal to the mass of the subject. Uncompressed leg length was l_0 , which was equal to the subject's leg length. Hip position (x, y) was measured from the hip to the center of pressure under the foot, both of which were known directly from the measured experimental data. The GRF for each leg was F_{leg} , which was also measured directly. To convert the measured human data into the model's leg stiffness k , the effective stiffness at each instant was calculated using

$$k_{inst} = \frac{\|F_{leg}\|}{\Delta l}$$

where Δl was the stance leg displacement, which was the difference between the subject's leg length l_0 and the distance from the hip to the center of pressure under the foot. The stiffness was then averaged over the step to find a step stiffness k . The stiffness of each leg was held constant throughout a step and changed during that leg's swing phase. The touchdown angle α_{TD} was found by calculating the angle between the ground and the line from hip to center of pressure at heel contact. A step began at touchdown and ended when the foot lifted off the ground. This model can be mathematically described using a set of continuous-time differential equations [30].

2. In the code, five subjects were skipped "due to poor data," but I didn't find a statement or explanation about their removal in the manuscript. The manuscript should detail all the data taken (the number of subjects is actually 17, not 12) and then a clear explanation why some could not be used.

The decision to skip the five subjects was made prior to performing any system identification. Because we did not attempt system identification on them, we left the number of subjects at 12 in the manuscript and added the following explanation to the code:

The decision to skip these subjects was made prior performing any system identification. Subject 5 was skipped because they were unable to stay centered on the treadmill, resulting in poor step identification. In addition, it would not have been possible to reliably determine the center of pressure location under the stance foot, making it impossible to accurately determine key model parameters. Subjects 6-9 were skipped because they experienced a somewhat different protocol.

3. End of introduction: Could the authors be more specific by "...using the SLIP model and informed by the first method." The abstract has the same wording. It is unclear at this point what "informed by the first method" means.

We have modified the phrasing to “... that both encoded the linearized SLIP model and matched the first linear system as closely as possible.” in both the abstract and introduction.

4. Table 1: The variables do not match Figure 1 or the equations that reference the table. Could similar variable names be used for consistency? Please restate what each variable refers to. Did the subjects exhibit a wide range of behavior (e.g. what was the standard deviation)?

Sec. 2.2 and Fig. 1 have been updated to use the same variable names as the rest of the paper. The Table 1 caption now defines each of the variables. The standard deviation for each variable has been added into the table.

5. Line 256 indicates that outliers were removed with `rmoutliers`. What was the criteria for removing the outliers, and how many outliers were removed? Since the summary statistics referred to human data, should the statement be included as part of Section 2.1?

We used the default threshold of three scaled median absolute deviations. This is now stated in the paper. We also give the percentage of steps removed. The paper now includes

Outliers were defined using the default threshold of three scaled median absolute deviations. For the linearized system, less than 5% of the test data were removed. For the optimized system, approximately 18% of the test data were removed when testing for accuracy, and approximately 8% were removed when evaluating the simulation results. For SINDY, approximately 8% of the test data were removed when testing for accuracy, and approximately 2% were removed when evaluating the simulation results.

This outlier detection removed data after using the identified system models, so we kept the explanation in the same place in the paper. Instead, we modified the phrasing at the beginning of the paragraph to make it more clear what data was being tested: “*Before calculating summary statistics, all of the test results from all subjects were grouped together and outliers were removed using the Matlab function `rmoutliers`.*”

6. I suggest ending the manuscript with a broader statement or two on the implications of the work, especially since the finding (linear performed best over nonlinear) was interesting and surprising.

We now end the paper with “*These results may indicate that the central nervous system uses an approximately linear controller when reacting to small perturbations caused by noise in the neuromuscular system, as opposed to the nonlinear controller that is often assumed.*”

We also added a similar statement to the end of the abstract: “*It may also suggest that human control of normal, unperturbed walking is approximately linear.*”

7. Supplementary data: the leg lengths for some of the subjects are zero. I assume this is related to removal of subjects.

Yes, this is correct. Because data for those subjects was never processed, their leg lengths were not included.

Reviewer 2 Comments

As per the email from Lianne Parkhouse on October 14, we have focused our changes on Reviewer 1's comments and have made very minimal changes in response to Reviewer 2's comments.

The author(s) has presented work on Comparing System Identification Techniques for Identifying 2 Human-like Walking Controllers. The paper has designed SLIP similar controller using human data for gait stability and controllability. However, the approach is simple and straightforward. The methodology and results are partially justified. The major rewrite of paper is required. The author(s) should consider the following review and revised fully.

1. The abstract needs to be written again as it are not clearly explaining the work done

We are unclear what aspect of the abstract the reviewer finds confusing. Given that neither Reviewer 1 nor the editor requested we make edits of this nature, we have not made changes in response to this comment.

2. The conclusion needs to be aligned with the abstract and future work should be more comprehensive.

We are unsure what aspect of the conclusion the reviewer feels is not aligned with the abstract. Given that neither Reviewer 1 nor the editor requested we make edits of this nature, we have not made changes in response to this comment.

3. The title is not matching with the proposed methodology. The title is not suitable rephrase it or change. The question mark cannot be title.

We chose to keep the title since we feel it accurately represents the work and neither Reviewer 1 nor the editor requested that we change it. The title is not a question, so we are unsure what the third sentence refers to.

4. On page , RELATED WORKS, you have mentioned only a select few algorithms that were used in human gait and bipedal gait, but in reality, there are numerous Machine learning and Deep learning methods which have been applied in the Human activity and gait recognition space. I suggest going through the literature review once again. Example: "Deep ensemble learning approach for lower extremity activities recognition using wearable sensors" by Jain et al. refer this work.

The authors are unclear what the reviewer is asking for here since this paper does not use machine learning. While it is possible that the system identification could have been done using machine learning, we did not use this approach nor are we aware of any papers that do. The suggested reference discusses classification, which is a distinct research topic from system identification.

5. Clearly explain the methodology as to how data pre-processing is performed. Merely mentioning that it is performed is not sufficient.

The first paragraph of Sec. 2.2 now includes details on how we converted the human subject data into the model parameters. It now reads

The SLIP model contains a point mass at the hip supported by two legs of different stiffnesses (Figure 1). A brief overview of the model is given here; the reader is referred to [6,30] for detailed mathematical derivations. The hip had mass m , which was equal to the mass of the subject. Uncompressed leg length

was l_0 , which was equal to the subject's leg length. Hip position (x, y) was measured from the hip to the center of pressure under the foot, both of which were known directly from the measured experimental data. The GRF for each leg was F_{leg} , which was also measured directly. To convert the measured human data into the model's leg stiffness k , the effective stiffness at each instant was calculated using

$$k_{inst} = \frac{\|F_{leg}\|}{\Delta l}$$

where Δl was the stance leg displacement, which was the difference between the subject's leg length l_0 and the distance from the hip to the center of pressure under the foot. The stiffness was then averaged over the step to find a step stiffness k . The stiffness of each leg was held constant throughout a step and changed during that leg's swing phase. The touchdown angle α_{TD} was found by calculating the angle between the ground and the line from hip to center of pressure at heel contact. A step began at touchdown and ended when the foot lifted off the ground. This model can be mathematically described using a set of continuous-time differential equations [30].

6. I didn't see any mathematical expressions and theoretical background of the proposed methodology. Only modifying the Deep learning model does not prove the proposed method's novelty.

The authors are unclear what this comment is referring to. This work did not use deep learning, nor do we claim to have developed a novel method. Instead, we apply existing methods to develop models that describe the human neuromuscular response, including its response to the noise inherent in the system.

7. A very vague description is provided for the model description. I suggest backing your statements with some results. Must provide the depth literature review, back ground support and application of proposed work.

The authors are unclear which model the reviewer is referring to. The SLIP model is a standard model in biped gait analysis, so the authors do not feel that including a detailed mathematical derivation is appropriate here as it is not a contribution of the paper. Instead, we have added additional citations to papers providing the detailed mathematical derivation, along with verbiage explicitly referring the reader to these earlier works. The beginning of Sec. 2.2 now includes "A brief overview of the model is given here; the reader is referred to [6,30] for detailed mathematical derivations."

If instead the reviewer is referring to the identified system models, the mathematical equations describing them are given in Sec. 2. The equations for the linear model are given in Eqs. 4 and 6. This is now more explicitly stated in the paper near the end of Sec. 2.3: "Thus, the linearized model and controller were given by Equations 4 and 6." The equations for the optimized model are given in Eqs. 9 and 10. This is now explicitly stated at the end of Sec. 2.4: "Thus, the optimized model and controller were given by Equations 9 and 10 with the terms calculated using the solutions to Equations 12 and 13." The equations for the SINDY model are given in Eqs. 14 and 15. This is now explicitly stated at the end of the 3rd paragraph in Sec. 2.5: "Thus, the SINDY model and controller were given by Equations 14 and 15 with J_S and K_S found using the iterative procedure just described." The complete models with numerical values for all terms are given in the supplemental material.

8. Proofreading of the paper is required. There are lots of grammatical mistakes which need to be addressed.

The authors have very carefully proofread the paper.

9. Must explain the motivation and feasibility in term of cost and applicability.

The authors are unclear exactly what the reviewer is requesting here.

10. Must compare with state of art work. Author can utilize ELM based classification technique and show the performance in table. Refer and cite this work: Patil, Prithvi, et al. "Clinical human gait classification: extreme learning machine approach." 2019 1st international conference on advances in science, engineering and robotics technology (ICASERT). IEEE, 2019.

The authors are unclear exactly what the reviewer is asking for here since the referenced paper focuses on classification while this paper focuses on system identification. It does not seem possible or helpful to compare the two works directly.

12. The workflow diagram is vague can design again.

The authors apologize, but we are not sure which figure the reviewer is referring to. None of the figures in the paper summarize the workflow.

13- The author(s) has missed many latest work of gait in recent year. The work should be compared with following papers in literature review to support and prove your results superiority.

Semwal, Vijay Bhaskar, et al. "Design of vector field for different subphases of gait and regeneration of gait pattern." IEEE Transactions on Automation Science and Engineering 15.1 (2016): 104-110.

Nandi, Gora Chand, et al. "Modeling bipedal locomotion trajectories using hybrid automata." 2016 IEEE region 10 conference (TENCON). IEEE, 2016.

Gupta, Anjali, et al. "Multiple task human gait analysis and identification: ensemble learning approach." Emotion and information processing. Springer, Cham, 2020. 185-197.

Bijalwan, Vishwanath, et al. "Fusion of Multi-sensor based Biomechanical Gait Analysis using Vision and Wearable Sensor." IEEE Sensors Journal (2021).

Semwal, Vijay Bhaskar, et al. "Pattern identification of different human joints for different human walking styles using inertial measurement unit (IMU) sensor." Artificial Intelligence Review (2021): 1-21.

Patil, Prithvi, et al. "Clinical human gait classification: extreme learning machine approach." 2019 1st international conference on advances in science, engineering and robotics technology (ICASERT). IEEE, 2019.

The papers the reviewer suggests either focus on gait classification or on methods to describe and choose specific joint angles, neither of which are directly related to the work in our paper. To avoid potential confusion about what our contribution is, we have decided not to include the suggested references in our paper.